# Exploration of Applicability of Diatom Indices to Evaluate Water Ecosystem Quality in Tangwang River in Northeast China

Hao Xue [1,2], Lei Wang [2,3], Lingsong Zhang [1], Yeyao Wang [2], Fansheng Meng [1,*] and Min Xu [1,*]

1 Chinese Research Academy of Environmental Sciences, Beijing 100012, China
2 China National Environmental Monitoring Centre, Beijing 100012, China; yimi003@sina.com (L.W.)
3 Shaanxi Environmental Monitoring Center, Xi'an 710054, China
* Correspondence: mengfs@craes.org.cn (F.M.); renyumeiwen1987@163.com (M.X.)

**Abstract:** The diatom index has been widely used in the evaluation of water ecological quality, but the applicability of the diatom index often varies in different study areas. The accuracy of the evaluation results depends on the applicability of the diatom index, especially when it is not applied to the place where it is created. In order to screen out the diatom index suitable for the evaluation of the water ecological quality of Tangwang River in northeast China, and to identify the factors affecting the accuracy of the diatom index, the community structure and water environment characteristics of 24 sample sites were investigated in Tangwang River in August 2018, and 18 diatom indices were calculated. The discriminative ability of diatom indices was analyzed using the box plot method, and the factors affecting the accuracy of the diatom index were identified by combining Pearson and Spearman correlation analyses. The results show that the discriminability of the Biological Diatom Index (BDI), Specific Pollution Sensitivity Index (IPS), Idse Leclercq (IDSE), Indice Diatomique Artois Picardie (IDAP), Diatom Eutrophication Pollution Index (EPI-D), Trophic Index (Rott TI), European Economic Community Index (CEE), and Watanabe Index (WAT) was the strongest, which could reasonably distinguish the reference group from the lightly damaged group. In general, the water ecological condition of Tangwang River Basin is good in the wet season, and the water ecological quality of about 80% of the sample sites was "moderate" or better. The main factors affecting the evaluation accuracy of the diatom index in Tangwang River Basin are the correlation strength between the diatom index and habitat quality, organic pollution, and nutrients. The coverage of diatom index species had no significant effect on the accuracy of evaluation. In order to reasonably evaluate the aquatic ecological status, it is recommended to use the diatom index, which has a good correlation with the environmental factors in the study area, or to establish a new diatom index based on the diatom community and environmental factors in the study area.

**Keywords:** diatom index; water ecosystem quality; *Achnanthidium minutissimum*; accuracy of diatom index

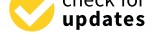



## 1. Introduction

The diatom generation time is one of the quickest among bioindicators of river water quality, and these algae divide frequently and can thus rapidly indicate a change in water quality [1]. Therefore, benthic diatoms have an unusual sensitivity to a variety of ecological conditions and have been widely used in river ecological quality monitoring [1,2]. At present, research on the application of the diatom index in river ecological and environmental quality assessment has been relatively mature, and dozens of river diatom indices have been developed, such as the Biological Diatom Index (BDI) [3] and Sladecek Saprobic Index (SLA) [4] developed in France, Trophic Diatom Index (TDI) [5] developed in the UK, Diatom Species Index for Australian Rivers (DSIAR) [6] developed in Australia, Diatom Pollution Tolerance Index (PTI) [7] developed in the USA, and Pampean Diatom Index (IDP) [8] developed in Argentina. These indices have been shown to be effective in identifying point

source pollution, organic enrichment, and eutrophication, and have been widely used [9]. In addition, some studies have modified and adjusted the existing diatom indices according to the characteristics of the study area. The South African Diatom Index (SADI) is based on the Specific Pollution Sensitivity Index (IPS) with the addition of endemic species from South Africa. Studies have shown that the SADI was effective in identifying the source of damage and determining the degree of impact [10], but the index still needs more in-depth regional research and verification [11].

In recent years, there have been more and more reports about the application of the diatom index in China. The diatom indices developed in France and Japan were used to evaluate the aquatic ecological status of the Pearl River Basin, and the results showed that it is feasible to widely apply diatom community organisms to monitor the ecological environment quality in southern China, but there are still some limitations [12]. The TDI was closely related to the changes in chemical oxygen demand (CODcr), total phosphorus (TP), and dissolved oxygen (DO) concentrations during a rainstorm event in the Songhua River Basin [13]. The IPS, BDI, and TDI have been proven to be significantly correlated with the water quality of the Ganhe River [14]. In addition, diatom indices have been used to construct the multimetric index (MMI) to evaluate the aquatic ecological status, and the benthic diatom index of biotic integrity (BD-IBI), based on the diatom index and taxonomic parameters, revealed the main reasons for the degradation of the ecosystem in the Hanjiang River [15]. In our previous study, we also evaluated the aquatic ecological status of the Wutong River Basin using the BD-IBI constructed based on the BDI, DSIAR, and other biological indicators [16]. Numerous studies have shown that it is feasible to widely use the diatom index for water ecological health assessment in China [12], but it is worth noting that the diatom index sometimes shows different adaptability in different regions [17,18]. The diatom index will show different effects when used in different regions due to the differences in flora in different regions and environmental differences that change the response of species to water quality characteristics [19].

The Tangwang River is located in northeast China, with many national nature reserves, abundant forest resources, a low intensity of human activities, and good natural conditions of the river. The study of diatom communities in the Tangwang River Basin can scientifically reflect the water's ecological conditions under minimal human activity intensity. In this study, based on the survey data of water quality, habitat, and diatoms in the Tangwang River in the wet season, we studied the applicability of 18 diatom indices in Tangwang River by using principal component analysis (PCA), box plot analysis, and Spearman's correlation analysis. The objective of this study is to identify a suitable diatom index for water ecological assessment in the Tangwang River Basin, accurately evaluate the water ecological environment quality of the basin, and determine the key factors influencing the applicability of the diatom index.

## 2. Research Methods

### 2.1. Study Area and Setting of Sampling Sites

The Tangwang River is an important tributary of the lower reaches of the Songhua River, with a total river length of 509 km, a watershed area of 21,245 km$^2$, an average annual runoff of $55.2 \times 10^9$ m$^3$, and an annual rainfall of about 610 mm. The average annual temperature of the Tangwang River Basin is about 1 °C, the annual average minimum temperature is $-22.6$ °C (January), and the annual average maximum temperature is 20.8 °C (July). The Tangwang River Basin is a typical mountainous forest watershed, which is one of the main forest areas in China. The terrain is high in the north and low in the south. The main landform types are low mountains, hills, and valleys. The river system is dendritic, with more than 600 tributaries, of which 6 tributaries have a catchment area of more than 1000 km$^2$. In this study, a total of 24 sampling sites (Figure 1) were set up and the main stream of Tangwang River and its main tributaries were sampled and investigated during the wet season (August) in 2018. In 2018, the average temperature was 2.2 °C and the rainfall was 903 mm.

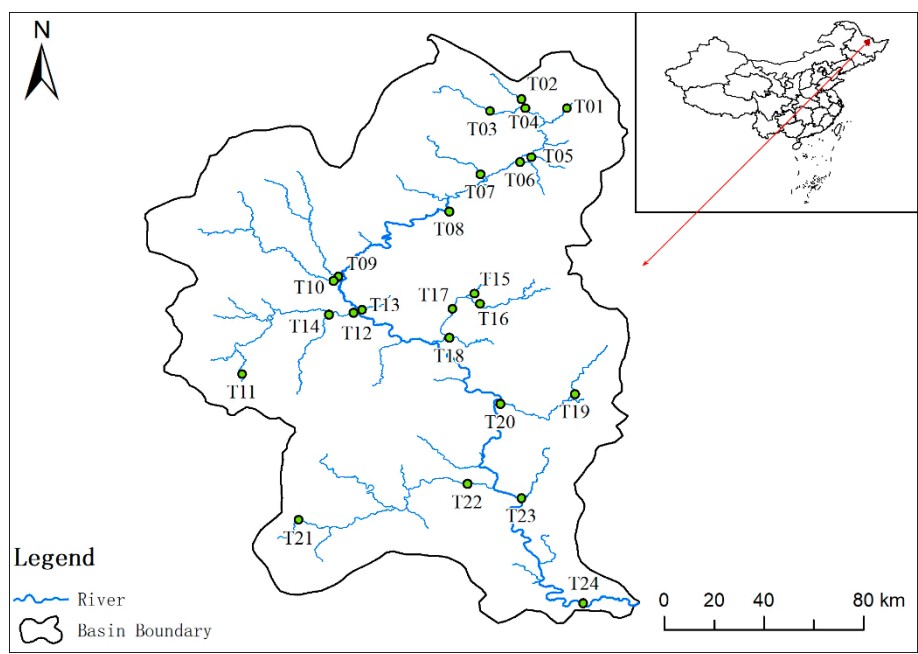

**Figure 1.** Sampling sites in Tangwang River.

### 2.2. Collection, Processing, and Analysis of Samples of Epiphytic Algae

Three stones were selected from different habitat conditions within 100 m upstream and downstream of the river sampling site (surface area on the stone < 200 cm$^2$). The attached algae within a circle of 2.8 cm in diameter on the surface of the stone were scraped with a stiff brush and washed into a stainless steel tray with purified water. A total of 5% formaldehyde solution was used as a fixative and added to the algae sample, which was stored in a wide-mouthed plastic bottle as a quantitative sample of the attached algae. For sites without stones, samples of attached algae on substrates such as dead branches and fallen leaves were brushed [16].

In the laboratory, part of the sample was acidified (concentrated nitric acid and concentrated sulfuric acid), and then diatom slides were made, identified, and counted under a 1000-fold optical microscope (Olympus BX51, Tokyo, Japan). At least 400 diatom cells were observed on each slide [16], and all diatom samples were identified as species.

### 2.3. Methods for Measuring Water Quality and Habitat Indicators

DO, electrical conductivity (EC), and pH value were determined in situ using a portable water quality analyzer (YSI Professional Plus, Yellow Springs, OH, USA). Water quality samples were collected at each sampling site and brought back to the laboratory after on-site pretreatment. $COD_{Cr}$ and permanganate index ($COD_{Mn}$) were determined by titration method, and ammonia nitrogen ($NH_4^+$-N), nitrate nitrogen ($NO_3^-$-N), total nitrogen (TN), and total phosphorus (TP) were determined by ultraviolet–visible spectrophotometer (MAPADA UV-1100, Shanghai, China). The qualitative habitat evaluation index (QHEI) and bottom characteristics (Bott.) were obtained according to the field scoring of river habitat evaluation index and evaluation standards established in the Liaohe River Basin in northeast China [20]. The QHEI integrated 10 indexes, including substrate, habitat complexity, velocity–depth combination, bank stability, channel alteration, stream flow conditions, vegetation diversity, water quality conditions, intensity of human activities, and riverside land use, with a full score of 20 for each index and a total score of 200.

### 2.4. Data Analysis

#### 2.4.1. Diatom Community Analysis

The Shannon diversity index (H′) is a comprehensive species richness index and a measure of the distribution uniformity for individuals that reflects the degree of community

complexity and the stability of community structures. Pielou evenness index (J) reflects the distribution evenness of individuals among species [13].

$$\mathrm{H}' = -\sum (n_i/N) \times \ln(n_i/N)$$

$$\mathrm{J} = \mathrm{H}'/\ln S$$

where $n_i$ and $N$ represent the number of individuals representing species $i$ and the total number of all species, respectively, and $S$ represents the number of species.

Data transformation [lg (x + 1)] was applied to the data of habitat quality, the relative abundance of attached diatoms, and the physical and chemical data of the water except pH, and PCA was used to determine the main environmental factors affecting the environmental quality of Tangwang River.

### 2.4.2. Screening Method for Reference and Damaged Sites

At present, there is no unified method for selecting reference sites [21], and the standard of reference conditions will vary according to the differences in water quality, topography, climate, soil, vegetation, and land use in different study areas [22], but minimum exposure to human activities is a criterion that needs to be strictly followed in the selection of reference sites [23]. Therefore, the score of human activity intensity and the score of riparian land use type is also used as one of the criteria for the selection of reference sites and damaged sites. In this study, we selected the site with higher vegetation cover, fewer human activities, and better water quality as the final reference site [24].

The content of humus in the soil of the Tangwang River Basin is relatively high, and the freeze–thaw cycle destroys the stability of soil aggregates, which is more likely to cause the loss of terrigenous organic matter into the river [25,26]. Accordingly, the content of humus in the river water is relatively high. Humus, as a macromolecular organic substance, will consume a large amount of oxygen and produce a large number of intermediate by-products under high temperature and strong acid conditions and will be detected with high values of $COD_{Cr}$, $COD_{Mn}$, and $NH_4^+$-N under laboratory conditions, so the background value of $COD_{Mn}$ in Tangwang River is high. The $COD_{Mn}$ value of all sampling sites in Tangwang River does not meet the Class III standard specified in the Environmental Quality Standards for Surface Water (GB3838-2002) [27]; therefore, in this study, without considering the $COD_{Mn}$ value, the reference site should simultaneously exceed the water quality standards of Class III standard and achieve a QHEI score higher than 160 points, while the damaged site should fail to meet the water quality standards of Class IV standard and obtain a QHEI score lower than 120 points.

### 2.4.3. Evaluation Based on Diatom Index and Correlation Analysis with Environmental Factors

Eighteen diatom indices were calculated by Omnidia 6.0 (IRSTEA Bordeaux, Roubaix, France) to evaluate the aquatic ecological health of Tangwang River Basin, which were BDI, IPS, TDI, SLA, IDP, Generic Diatomic Index (IDG), Descy Index (Descy), Idse Leclercq (IDSE), Indice Diatomique Artois Picardie (IDAP), Diatom Eutrophication Pollution Index (EPI-D), European Economic Community Index (CEE), Watanabe Index (WAT), Lobo Index (Lobo), Hurlimann Trophic Index (DI-CH), Trophic Index (Rott TI), Saprobic Index (Rott SI), Trophic Diatom Index for Lakes (TDIL), and Steinberg and Schiefele Index (SHE). An introduction to these 18 diatom indexes can be found on Omnidia's official website (https://omnidia.fr/en/, accessed on 10 October 2023). The discriminant ability of the diatom index in the Tangwang River Basin was judged by the box diagram method; a high score indicates that the diatom index can effectively distinguish between different ecological health conditions [28]. Spearman correlation analysis was carried out on 18 diatom indices and 10 environmental factors, and the correlation coefficient between each index and environmental factors was calculated.

2.4.4. Factors Affecting the Applicability of Diatom Indices

In this study, the concept of "accuracy of diatom index evaluation (accuracy)" was proposed to analyze the factors affecting the applicability of diatom index. The accuracy is calculated by the correct recognition rate of the diatom index to the reference sites and the damaged sites. In the first step, taking BDI as an example, a rank score of 1 to 24 to each sampling site was assigned in turn according to the BDI score from low to high, then, respectively, the sum of the rank scores of the reference site and the damaged site was calculated, and the above calculation process was carried out on the rest 17 diatom indices. In the second step, the sum of the rank scores of the reference sites of the 18 diatom indices was ranked from low to high and a score of 1 to 18 was given in turn, which was recorded as $a$, and then the sum of the rank scores of the damaged sites of the diatom indices was ranked from high to low and a score of 1 to 18 was given in turn, which was recorded as $b$.

$$\text{accuracy} = \frac{a+b}{2}$$

The higher the accuracy score, the stronger the ability of the diatom index to correctly identify the reference site and the damaged site.

Pearson correlation analysis was used to analyze the correlation between various influencing factors and the accuracy of diatom index evaluation, including the strength of the correlation between diatom index and water environment factors as well as QHEI, and the coverage of diatom index species (the proportion of diatom species involved in the calculation of diatom index to the total number of species at the site, %).

In this study, all analyses were conducted through the R 3.5.2 (https://www.r-project.org/, accessed on 10 October 2023), an open source software developed by Ross Ihaka and Robert Gentleman. PCA analysis, Pearson correlation analysis, and Spearman correlation analysis were conducted through the "vegan 2.6-4" package, and box plot analysis was conducted through the "ggplot2 3.4.4" package.

## 3. Results and Discussion

### 3.1. Diatom Community Structure

A total of 113 species of periphytic algae belonging to 6 phyla and 49 genera were identified in Tangwang River. Among them, 99 species belonged to *Bacillariophyta*, 6 species to *Chlorophyta*, 5 species to *Cyanophyta*, and 1 species to *Euglenophyta*, *Cryptophyta*, and *Xanthophyta*, respectively. Species belonging to *Navicula*, *Nitzschia*, and *Hemipolaris* were the most numerous, with 19, 13, and 10 species, respectively. The number of benthic diatom species identified at the T04 site was the largest, with a total of 30 species; the number of benthic diatom species identified at the T12 site was the least, with a total of 11 species; the average number of benthos species at all sites in Tangwang River was 20; and there was no significant difference in the number of species among sites.

In the wet season, the average score of the Shannon diversity index of benthic diatoms in Tangwang River was 2.81, and the average score of the Pielou evenness index was 0.65. The Shannon diversity index score of T04 was the highest (4.29), and the Pielou evenness index score of T24 was the highest (0.91). The Shannon diversity index and Pielou evenness index of T09 were the lowest, which were 0.96 and 0.25, respectively. The main reason for the low species evenness at T09 is that the relative abundance of *Achnanthidium minutissimum* at T09 was close to 90%. *A. minutissimum* is widely distributed all over the world and is a common species in many regions [29]. *A. minutissimum* has a Mcnaughton dominance of 0.32, which was the dominant species in Tangwang River, which may be due to the large amount of water in Tangwang River in the wet season. Studies have shown that *A. minutissimum* usually reaches a higher dominance when the amount of water is large [30].

### 3.2. Environmental Factor Analysis

According to the screening criteria of reference and damaged sites, the 24 sampling sites were divided into three groups: group 1 was the reference group (G1) with 6 sites, group 2 was the lightly damaged group (G2) with 12 sites, and group 3 was the damaged group (G3) with 6 sites, and the QHEI and main physicochemical parameters of each group are shown in Table 1.

**Table 1.** State of QHEI and water quality between reference and impaired sites (mean $\pm$ SD).

| Group | QHEI | pH | $\rho$/(mg/L) | | | | | |
|---|---|---|---|---|---|---|---|---|
| | | | $COD_{Cr}$ | $COD_{Mn}$ | DO | TN | $NH_4^+$-N | TP |
| G1 | 165.33 $\pm$ 5.28 | 6.78~7.76 | 19.16 $\pm$ 4.71 | 9.89 $\pm$ 3.41 | 11.12 $\pm$ 0.59 | 0.83 $\pm$ 0.10 | 0.37 $\pm$ 0.19 | 0.00 $\pm$ 0.00 |
| G2 | 135.73 $\pm$ 12.94 | 6.46~8.66 | 23.05 $\pm$ 4.73 | 12.10 $\pm$ 3.37 | 9.69 $\pm$ 1.24 | 1.01 $\pm$ 0.19 | 0.50 $\pm$ 0.20 | 0.01 $\pm$ 0.01 |
| G3 | 112.5 $\pm$ 10.21 | 6.77~7.83 | 23.65 $\pm$ 4.26 | 11.04 $\pm$ 3.66 | 9.10 $\pm$ 1.06 | 1.47 $\pm$ 0.21 | 0.78 $\pm$ 0.34 | 0.02 $\pm$ 0.01 |

The Tangwang River Basin is characterized by a low intensity of human activities and good habitat quality. The headwater areas of some tributaries are almost in a natural state, so they are classified into reference groups. The intensity of human activities increased in the surrounding areas of Yichun City and the lower reaches of the mainstream, the water quality and habitat quality became worse, and some sites were classified as damaged groups. The main over-standard factors of the water environment in Tangwang River were $COD_{Mn}$ and TN, and the $COD_{Mn}$ at all sites was high, which might be caused by the high content of organic matter and humus in the soil of the Songhua River Basin. Sampling sites with a substandard TN concentration were mainly concentrated downstream of the main stream and the confluence areas of some tributaries, and the increase in TN concentration may be caused by human activities such as agricultural planting.

PCA analysis was performed on 11 environmental factors, and the results showed that the interpretation rates of the first and second principal components were 0.35 and 0.22, respectively, and the cumulative interpretation rates of the first and second principal components were close to 60%, so the ranking diagram (Figure 2) was drawn with the first and second principal component axes. The circle in Figure 2 represents the average contribution rates of each parameter. The PCA results show that the contribution rates of seven environmental factors ($NH_4^+$-N, $NO_3^-$-N, TN, DO, QHEI, $COD_{Cr}$, and $COD_{Mn}$) were higher than the average contribution of all factors.

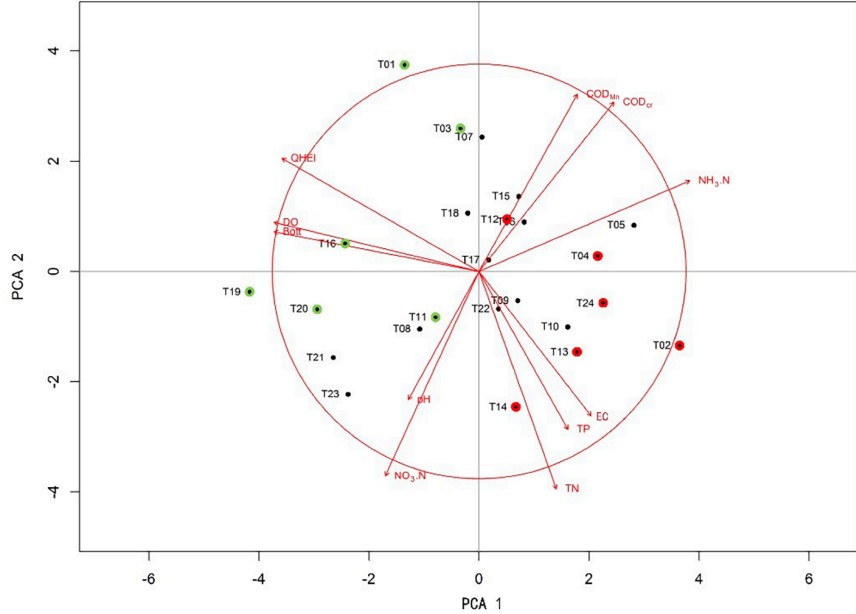

**Figure 2.** PCA ordination diagram of the Tangwang River environmental data.

Reference sites (green sites in Figure 2) are mainly located on the left side of the ordination axis, indicating that the QHEI and DO concentrations at these sites are higher and the substrate condition is better. T21 and T23, which were also located on the left side of the ranking axis, were not included in the reference group because they were located on the lower left side of the ranking axis, indicating that the concentrations of nitrate nitrogen and total nitrogen at these two sites were high. Damaged sites (red sites in Figure 2) are mainly located on the right side of the ordination axis, indicating that the habitat quality of this part of the site is poor, and the nutrient concentration (TN, $NH_4^+$-N, TP) is high. Some reference sites, such as T01 and T03, are located at positions with a high $COD_{Mn}$ and $COD_{Cr}$ value, which just shows that, in the Tangwang River Basin, due to the influence of background value factors, some areas with a good habitat and almost no human activities have a high $COD_{Mn}$ and $COD_{Cr}$ value.

### 3.3. Analysis of Diatom Index Evaluation Results

The discriminant ability of the Shannon diversity index, Pielou evenness index, and 18 diatom indices was analyzed by the box plot method. The results (Figure 3) showed that the box plot scores of the Shannon diversity index and Pielou evenness index were low, indicating that their discrimination ability was poor and that they could not reasonably distinguish the reference group from the damaged group. Similarly, the correlation between alpha diversity and water environmental quality in the Songhua River Basin during the rainstorm period is weaker than the TDI [13]. Indices based on species composition, particularly the diatom trophic index, are more closely linked to nutrients than biomass or diversity of species [31]. This may be because, in the process of community succession, the increase in species diversity is usually a benign adaptation of the ecosystem to mild external disturbance. A large number of studies [32–34] have shown that in the process of community succession, species diversity will increase first and then decrease with the increase in external disturbance intensity; that is, a hump-shaped effect. The diatom species diversity in Tangwang River did not show a downward trend with the external disturbance, so it can be inferred that the Tangwang River basin was affected by a low external disturbance intensity, and the overall environmental quality was good and did not reach the critical value of the hump effect. In addition, there is evidence that biofilm structures and extracellular polymeric substances have protective effects, and that the ability of ecological state classification indicators based on diatom diversity and abundance to distinguish environmental impacts is largely limited [9].

Among the 18 diatom indices, Lobo and the TDI had the worst discriminatory ability and the lowest box plot scores for the reference and damaged groups and could not effectively distinguish the reference group from the lightly damaged group. The remaining 16 diatom indices could reasonably distinguish the reference group from the damaged group, among which, the BDI, IPS, IDSE, IDAP, EPI-D, Rott TI, CEE, and WAT showed the strongest discriminant ability, and the box plot scores between the reference group and the mild damaged group were ≥2, which indicates that the reference group and the mild damaged group can be effectively distinguished by the above diatom indices.

The BDI and IPS, both established in France, are widely used diatom indices worldwide, showing strong adaptability in Europe [35], Africa [36], and Southeast Asia [37]. A wide range of species and high accuracy are the common characteristics of the BDI and IPS. The IDSE, also established in France, can effectively indicate the saprophytic degree and eutrophication degree of the water. The relevant study of the Chambal River in India [38] shows that IDSE is significantly correlated with environmental factors such as DO, $COD_{Cr}$, and biochemical oxygen demand, and IDSE can well predict the change in water environmental quality of the Chambal River.

Based on the health status classification standard [39] used by Eloranta et al., the health status of all sampling sites was evaluated according to the scoring results of the above eight diatom indices. The evaluation results are shown in Figure 4. The results show that the BDI and WAT had the highest scores among the eight diatom indices, and 50% of the sampling

sites were evaluated as "high". The Rott TI had the lowest score, there were no "high" and "good" sites in the evaluation results, and nearly 50% of the sites were evaluated as "bad". The results of the IPS and CEE are the most evenly distributed, including five types of evaluation results from "high" to "bad". In general, the water ecological health of Tangwang River Basin in the wet season was good, and the water ecological health of about 80% of the sites was "moderate" or better.

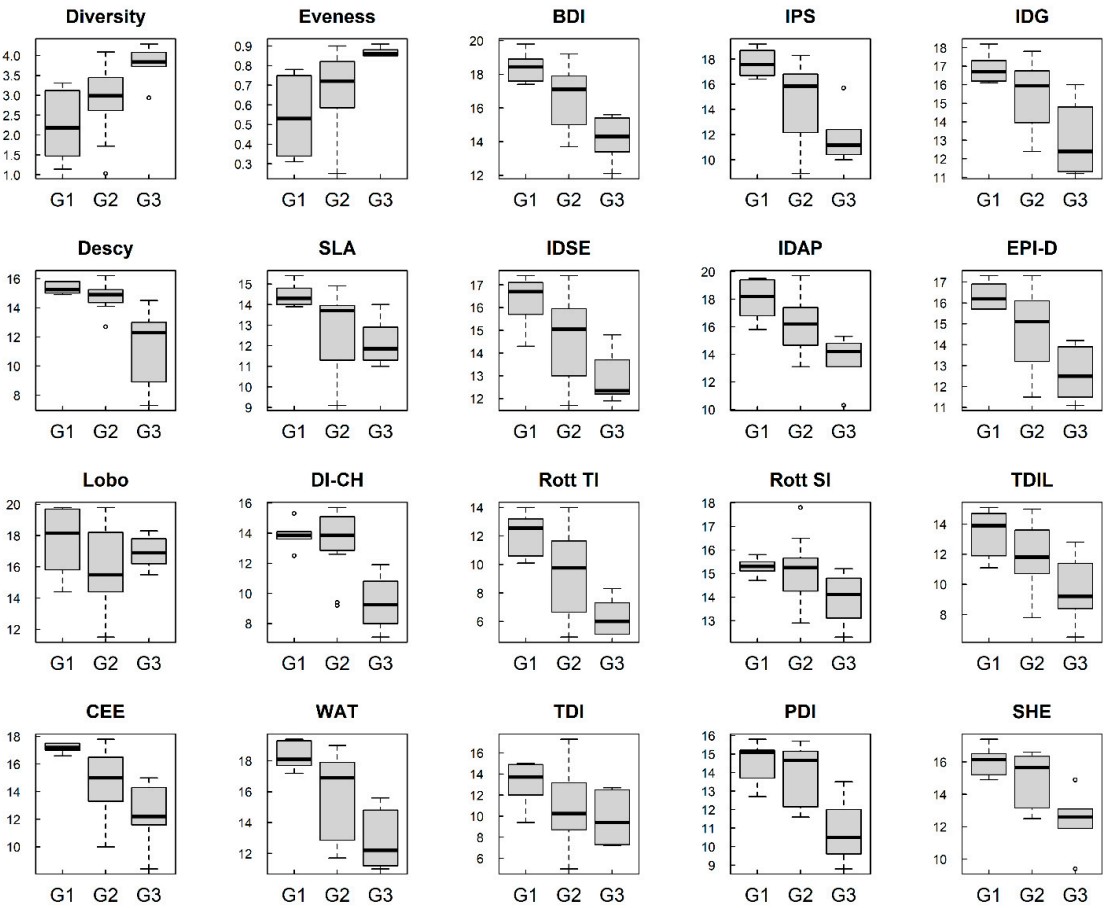

**Figure 3.** Boxplot analysis of diatom indices.

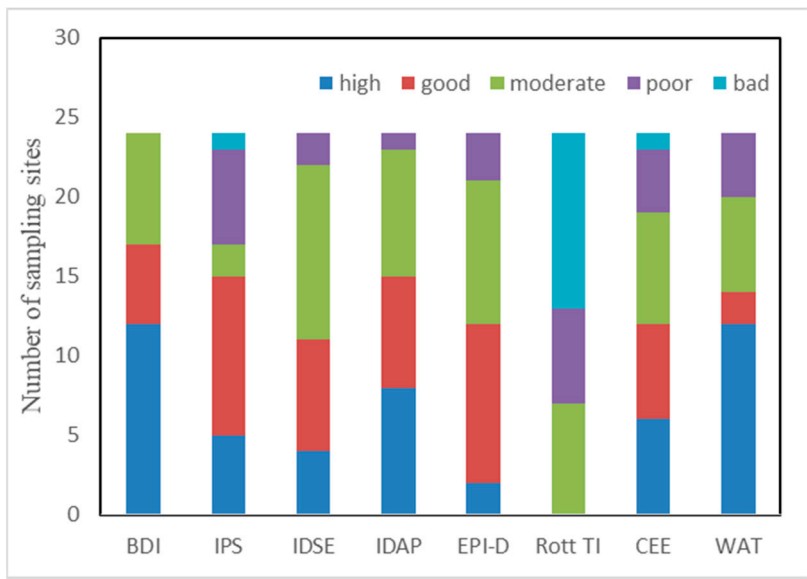

**Figure 4.** Evaluation results of the 8 diatom indexes showing the strongest discrimination ability.

### 3.4. Correlation Analysis of Diatom Index and Environmental Factors

Correlations between diatom indices and environmental factors were calculated using Spearman correlation analysis, and the results (Table 2) show that, among 11 environmental indices, EC, pH, $COD_{Cr}$, $COD_{Mn}$, and $NO_3^-$-N were not significantly correlated with all diatom indices, DO and $NH_4^+$-N were significantly correlated with only a few diatom indices, TP and TN were significantly correlated with about 50% of diatom indices, and the QHEI was significantly correlated with all of the 18 diatom indices. This indicates that the majority of the diatom indices selected can effectively indicate the habitat health status of Tangwang River, while approximately half of these indices can reasonably reflect the nutrient concentration levels in the river. The IPS, SLA, IDSE, EPI-D, CEE, and WAT were significantly correlated with four environmental factors; Descy, Lobo, and the TDI were the least correlated with environmental factors and only significantly correlated with the QHEI; the remaining diatom indices were significantly correlated with two or three environmental factors.

**Table 2.** Correlation coefficients between the values of diatom indices and the results of water quality.

| Index | DO | EC | pH | $COD_{Cr}$ | $COD_{Mn}$ | $NH_4^+$-N | $NO_3^-$-N | TN | TP | QHEI |
|---|---|---|---|---|---|---|---|---|---|---|
| BDI | 0.35 | −0.17 | −0.04 | −0.21 | −0.03 | −0.37 | −0.11 | −0.46 * | −0.40 | 0.80 ** |
| IPS | 0.54 * | −0.32 | −0.02 | −0.17 | 0.01 | −0.49 * | 0.03 | −0.35 | −0.57 ** | 0.80 ** |
| IDG | 0.33 | −0.25 | 0.03 | −0.15 | 0.04 | −0.38 | −0.13 | −0.44 * | −0.44 * | 0.71 ** |
| Descy | 0.41 | −0.14 | 0.03 | −0.25 | −0.07 | −0.40 | −0.07 | −0.40 | −0.39 | 0.87 ** |
| SLA | 0.56 ** | −0.35 | −0.08 | −0.18 | −0.03 | −0.52 * | 0.15 | −0.21 | −0.56 ** | 0.64 ** |
| IDSE | 0.42 * | −0.25 | −0.05 | −0.12 | 0.06 | −0.36 | −0.02 | −0.42 * | −0.51 * | 0.82 ** |
| IDAP | 0.40 | −0.20 | −0.10 | −0.21 | −0.03 | −0.35 | −0.14 | −0.55 * | −0.49 * | 0.82 ** |
| EPI-D | 0.45 * | −0.26 | −0.11 | −0.23 | −0.04 | −0.39 | −0.07 | −0.47 * | −0.50 * | 0.80 ** |
| Lobo | 0.26 | −0.19 | −0.03 | 0.11 | 0.19 | −0.09 | 0.05 | −0.15 | −0.23 | 0.42 * |
| DI-CH | −0.01 | 0.07 | 0.12 | −0.16 | −0.03 | −0.10 | −0.31 | −0.55 * | −0.13 | 0.66 ** |
| Rott TI | 0.40 | −0.25 | 0.00 | −0.03 | 0.14 | −0.37 | −0.14 | −0.54 * | −0.44 * | 0.81 ** |
| Rott SI | 0.09 | −0.14 | −0.11 | −0.03 | 0.12 | −0.18 | −0.24 | −0.45 * | −0.26 | 0.53 * |
| TDIL | 0.14 | −0.10 | 0.01 | 0.04 | 0.19 | −0.14 | −0.21 | −0.56 ** | −0.34 | 0.71 ** |
| CEE | 0.52 * | −0.35 | 0.01 | −0.29 | −0.10 | −0.51 * | 0.04 | −0.36 | −0.54 * | 0.83 ** |
| WAT | 0.48 * | −0.36 | −0.03 | −0.11 | 0.08 | −0.48 * | −0.04 | −0.41 | −0.56 ** | 0.72 ** |
| TDI | 0.34 | −0.28 | −0.12 | −0.01 | 0.17 | −0.37 | 0.09 | −0.29 | −0.30 | 0.68 ** |
| PDI | 0.16 | −0.08 | 0.04 | 0.06 | 0.22 | −0.24 | −0.27 | −0.59 ** | −0.34 | 0.67 ** |
| SHE | 0.16 | −0.11 | −0.01 | −0.05 | 0.10 | −0.18 | −0.37 | −0.62 ** | −0.48 * | 0.64 ** |

Note: ** means extremely significant correlation ($p < 0.01$), * means significant correlation ($p < 0.05$).

Numerous studies have shown that the relationships between different diatom indices and environmental variables vary widely from very weak to strong in different regions, which may be due to the different environmental factors or nutrient gradients used in the establishment of each index. The TDIL was highly correlated with TP in the relevant study of Irish lakes [40], the PDI also showed different correlations with TP concentrations in different areas in the evaluation of Lake Erie, and PDI scores were not significant in predicting TP in the western basin [17]. In this study, the low correlation between the TDIL, PDI, and TP may be due to the low concentration of TP in the Tangwang River Basin, which is much lower than the reference value of TP when the TDIL and PDI were established [41]. In addition, diatom communities are also affected by gradients of natural factors unrelated to nutrient concentrations, especially flow velocity, turbidity, and alkalinity [31].

### 3.5. Impact Analysis of Diatom Index Evaluation

The correlation between different influencing factors and the accuracy of diatom index evaluation was calculated using Pearson correlation analysis. The circles in the matrix of Figure 5 represent the strength of the correlation, and the larger the circle and the darker the color, the greater the correlation coefficient. The results show that the larger the correlation coefficient between the accuracy of diatom index evaluation and the QHEI, the

higher the accuracy of diatom index evaluation, and the correlation between them was highly significant ($R^2 = 0.81$, $p < 0.01$). The accuracy of diatom index evaluation was also significantly and positively correlated with $COD_{Cr}$ ($R^2 = 0.65$, $p < 0.01$). In addition, the correlation coefficients of the diatom index with ammonia nitrogen and total phosphorus were also positively correlated with the accuracy of the diatom index evaluation ($R^2 > 0.55$, $p < 0.01$). The above factors represent the correlation between diatom index and habitat quality, organic pollution, and trophic status, respectively. Numerous studies have shown that the strength of the correlation with environmental factors is one of the main reasons for the efficiency of diatom index evaluation [42,43].

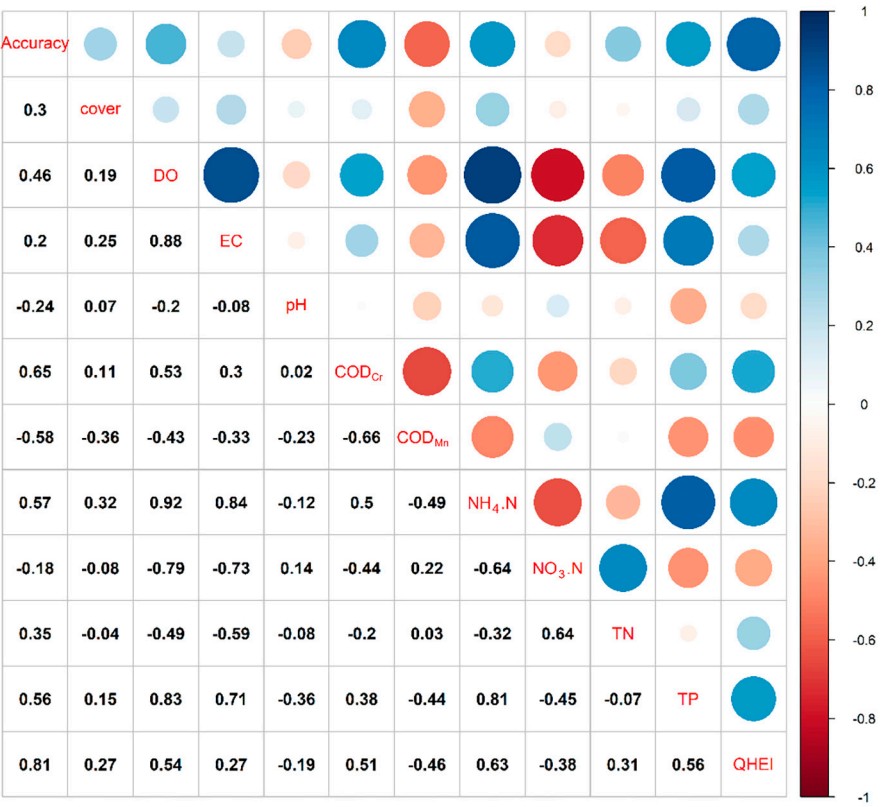

**Figure 5.** Analysis of influence on accuracy of diatom indices.

It is noteworthy that the stronger the correlation between the diatom index and $COD_{Mn}$ in this study, the lower the accuracy of diatom index evaluation, and the correlation between them was also very significant ($R^2 = -0.58$, $p < 0.01$). In the process of organic matter concentration determination, the oxidation rate of the acidic potassium dichromate method is higher than that of the potassium permanganate method, so $COD_{Mn}$ is usually used to indicate the concentration of organic matter in relatively clean surface water, and $COD_{Cr}$ is usually used to indicate the concentration of organic matter in wastewater. In the source area of Tangwang River, the degradation rate of litter by microorganisms is affected by the cold temperate climate, and the litter cannot be completely degraded, resulting in a large number of accumulations. In addition, the soil in the study area is rich in humus, and the concentration of humus in the river is high, so it is easy for the humus in the river to be oxidized when using the potassium permanganate method to determine the concentration of organic matter, which leads to the high $COD_{Mn}$ value in the source area of the river, which is less affected by human activities and cannot truly reflect the impact of human activities. Owing to the influence of human activities, the amount of wastewater flowing into the lower reaches of Tangwang River has increased, and the content of industrial organic matter and macromolecular organic matter in the river has increased. It is difficult for this part of organic matter to be oxidized when potassium permanganate is used for oxidation, but it is easier for it to be oxidized when potassium dichromate is used for

oxidation. Therefore, we infer that the $COD_{Cr}$ value may better reflect the concentration of industrial organic matter and macromolecular organic matter discharged into the river by human activities in the process of water quality measurement in the Tangwang River Basin; that is, the $COD_{Cr}$ value can better indicate the degree of organic pollution caused by human activities, which should be the main reason for the positive correlation between the accuracy of diatom index evaluation and $COD_{Cr}$ and a negative correlation with $COD_{Mn}$.

The species coverage of the diatom index may also affect the accuracy of the evaluation. Low coverage may have serious consequences because some key indicator groups may be ignored in the process of index calculation, resulting in the loss of important information and limiting the ability to evaluate the ecological integrity of communities [44]. Too few taxa used for diatom index calculation may be one of the factors leading to the poor performance of diatom index (such as WAT) evaluation [45]. It has also been suggested that too many taxonomic units in the diatom indices and the resulting taxonomic difficulties may in turn limit the application of the indices [46]. In this study, diatom index species coverage also affected the accuracy of diatom index evaluation, but it was not significant ($p > 0.05$). In the case of the WAT, although only 41.59% of diatom species were used to calculate the diatom index, much lower than the IDG (100%), the WAT performed slightly better than the IDG in Tongwang River, suggesting that perhaps the proportion of reliable taxonomic units is more determinant of the applicability of the index to the study area than species coverage [18]. Therefore, to improve the applicability of the diatom index, some scholars have adjusted and optimized the environmental tolerance of the species in the diatom index [47], used the multimetric index (MMI) index to assess the ecological quality [48], or created a new diatom index in the region [49]. However, the establishment of the diatom index requires a large amount of data accumulation, and the new diatom index also requires much verification and adjustment [10]. Therefore, in the absence of locally developed diatom indices, these mature diatom indices can be used for the daily evaluation of aquatic ecological conditions [43].

The error of species identification is also one of the potential factors affecting the accuracy of diatom index evaluation, such as *A. minutissimum*, the dominant species in this study. Many varieties are widely considered to be *A. minutissimum*, it is difficult to identify the differences between these varieties with standard microscopic methods, and sometimes the evaluation results are better after *A. minutissimum* is eliminated in the evaluation process [50]. To avoid the difference in evaluation results caused by the error of species identification, some scholars have tried to establish a non-classification evaluation method based on diatom community samples, which directly uses diatom molecular information rather than species information to evaluate river biomass [51].

In addition, some species, especially those widely distributed worldwide, may exhibit different tolerances in different regions, so using them to calculate the indices does not always give consistent results [52]. As the database covers many regions of the world, ISP is widely used and shows good consistency and efficiency in European countries [49] and in China [14], but, in some regions, satisfactory evaluation results are not obtained [42]. Some species with wide worldwide distribution, such as Nitzschia palea and *Gomphonema parvulum*, may be influenced not only by environmental factors but also by biogeographic or phylogeographic differences, based on which the correlation between index scores calculated and water quality variables differs [53]. Due to temporal and spatial differences in environmental factors, sometimes evaluation methods based on small areas are better than those based on large-scale research areas [47].

## 4. Conclusions

The diatom-based biotic indices showed better discriminatory ability compared with the diversity indices in the evaluation of the water ecological health of Tangwang River during the wet season. Among the 18 diatom indices evaluated, the reference group and the damaged group could be reasonably distinguished by diatom indices other than Lobo and the TDI. The BDI, IPS, IDSE, IDAP, EPI-D, Rott TI, CEE, and WAT have the strongest

discriminatory ability and can reasonably distinguish the reference group from the mildly damaged group. The IPS and CEE have the most evenly distributed evaluation results, including five categories from "high" to "bad", and the IPS and WAT showed a significant correlation with the concentration of TP in the river. Therefore, the IPS, CEE, and WAT should be the most suitable diatom indices for Tangwang River. In general, the water ecological health of the Tangwang River basin is good during the wet season, and about 80% of the sites have "moderate" or better water ecological health.

Both the correlation strength between the diatom index and environmental factors and the species coverage of the diatom index will affect the efficiency of water ecological evaluation. In this study, the primary factor affecting the accuracy of the diatom index in the aquatic ecological evaluation of the Tangwang River Basin is the correlation strength between the diatom index and habitat quality, organic pollution, and nutrients. The species coverage of the diatom index has no significant effect on the accuracy of evaluation. Therefore, in the process of water ecological status assessment, it is suggested to select the diatom index that has a good correlation with the environmental factors in the study area or to build a new diatom index based on the diatom community and environmental elements in the study area.

**Author Contributions:** Conceptualization, H.X.; data curation, H.X. and M.X.; formal analysis, H.X. and L.W.; funding acquisition, F.M.; investigation, H.X. and L.Z.; methodology, H.X.; project administration, Y.W., L.Z. and F.M.; resources, F.M.; software, H.X. and M.X.; supervision, Y.W. and L.Z.; validation, F.M.; visualization, H.X.; writing—original draft, H.X.; writing—review and editing, F.M. and M.X. All authors have read and agreed to the published version of the manuscript.

**Funding:** Joint research project II on ecological environment protection and restoration of the Yangtze River (2022-LHYJ-02-0506-09).

**Conflicts of Interest:** The authors declare no conflict of interest.

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
