# Peer review of "Exploration of Applicability of Diatom Indices to Evaluate Water Ecosystem Quality in Tangwang River in Northeast China"

_water, doi:10.3390/w15203695_

Round 1

Reviewer 1 Report (New Reviewer)

This is an interesting evaluation of the Application of multiple diatom indices and water quality data from a study region in Tangwang River catchment. It considers the correlation between the different indices and the water quality parameters.

There has clearly been a lot of background work completed to gather the dataset, and the authors should be commended for these efforts and commitment.

The paper evaluates 18 indices and appears to suggest that the 8 diatom-based biotic indices are more "accurate" than the others. Acknowledging that there might not be a single, clear winner from the group, perhaps the paper should provide a recommendation of the 3 best indices as part of the conclusions?

Figure 5 is unclear and some additional axis labels might help. Alternately, the correlation coefficients could be displayed inside the coloured circles, to align with the text in Lines 329-341.

The paper would also benefit from some images of the most prevalent diatom species recorded by the research, if available, near Lines 192 - 209.

Overall, a solid manuscript and interesting observations.

Author Response

Reviewer 2 Report (New Reviewer)

Comments to the manuscript: Exploration of applicability of diatom indices to evaluate water ecosystem quality in Tangwang River in northeast China.

The idea for the article is good but needs refinement. In particular, the methodology requires extensive changes. Also, the description of the result and discussion should be more profound.

l. 71-80 This part is not an Introduction. What's more, there is a lack of aim in the study. The introduction should also give readers information about diatoms and why they are so important from an ecological point of view.

l.112 add full names of all abbreviations. Add full names of abbreviations when they appear first time in the text, e.g. l.157

PCA - how was it done, and in what program? This should be included in the methodology

l.140-152 mix of results/discussion part with methodology. please, divide it into appropriate paragraphs

part 2.4.3 please provide the methodology of used indexes. It could be in the appendix.

l.186 open source software - provide website

l. 201 Shannon diversity index & Pielou evenness index - lack in the methodology! It's apparent first time in the result part.

part 3.2 adds to the methodology. Which sites were included in G1, G2, and G3?

Figure 4 is illegible. In particular, the legend does not contribute. Add colors.

part 3.4: what do significant high or low correlation coefficients indicate? Where does it come from? Please explain.

Author Response

Reviewer 3 Report (New Reviewer)

The article entitled 'Exploration of applicability of diatom indices to evaluate water ecosystem quality in Tangwang River in northeast China' presents the evaluation of the water quality of the Tangwang River using 18 diatom indices. Although the results are interesting from a scientific point of view, I think that some changes are necessary before it can be accepted for publication. The suggestions can be found in the attached document.

Round 2

Reviewer 2 Report (New Reviewer)

Dear Authors, I regret to say that you did not include all the comments I wrote about.
At work there is still a lack of, among others: purpose of the work - it has not been added, I asked for the full names of the abbreviations - number 171, where are they?
The changes introduced are cosmetic and do not take into account my comments or are even ignored.

Author Response

Response: Thank you for your valuable comments. We have added the purpose of the work in the last sentence of the introduction chapter. “The objective of this study is to identify a suitable diatom index for water ecological assessment in the Tangwang River Basin, accurately evaluate the water ecological environment quality of the basin, and determine the key factors influencing the applicability of the diatom index. “

The full names of the DO, CODCr, BDI, SLA, TDI, IDP, IPS, is divided into the following chapters.

“TDI was closely related to the changes in chemical oxygen demand (CODcr), total phosphorus (TP), and dissolved oxygen (DO) concentrations during a rainstorm event in the Songhua River Basin [13].”

“At present, research on the application of diatom index in river ecological and environmental quality assessment has been relatively mature, and dozens of river diatom indices have been developed, such as Biological Diatom Index (BDI) [3] and Sladecek Saprobic Index (SLA) [4] developed in France, Trophic Diatom Index (TDI) [5] developed in the UK, Diatom Species Index for Australian Rivers (DSIAR) [6] developed in Australia, Diatom Pollution Tolerance Index (PTI) [7] developed in the USA, Pampean Diatom Index (IDP) [8] developed in Argentina.”

“South African Diatom Index (SADI) is based on Specific Pollution Sensitivity Index (IPS) with the addition of endemic species from South Africa.”

Reviewer 3 Report (New Reviewer)

Accept in present form

Author Response

Thank you for your valuable comments, which are of great help to our manuscript improvement.

This manuscript is a resubmission of an earlier submission. The following is a list of the peer review reports and author responses from that submission.

Round 1

Reviewer 1 Report

General comments

The paper contains a very interesting comparison of a large set of diatom indexes, allowing for the assessment of the state of the aquatic environment, and an attempt to determine the most useful indices for a specific case of the Tangwang River in north-eastern China, flowing through relatively poorly urbanized areas. The site of study is undoubtedly very interesting, and the analyzes carried out competently. However, this is a single case study and only for one research season, hence the usefulness of the obtained results is somewhat limited. Nevertheless, within the adopted scope, the work is done at a good level. My objections mainly concern the chapters 'Material and methods' and 'Conclusions'.

1. Study area (lines 80-85) needs to be supplemented with climatic conditions (average annual temperatures, temperature of the coldest and hottest month of the year, annual precipitation). Due to the fact that the research was conducted in one season, meteorological data for the 2018 season should also be included here, with a reference to multi-year conditions. It is important whether the year of research was average or deviated from the average conditions, e.g. for the multi-year period 1991-2020. I am convinced that there are available data from weather stations in the study area.

2. It is advisable to supplement the 'Material and methods' with a more precise characterization of the studied river and its catchment area: topography, nature of the river, use of the catchment area, degree of urbanization. The Tangwang River seems to be an excellent object for such research - the upper part of the river is very natural, with a poorly urbanized natural catchment, and the lower course is more developed. It is worth describing it in the content of the chapter.

3. Subsection 2.3. (lines 101-110) needs to be supplemented with what laboratory methods were used to determine water quality indicators.

Also, the QHEI indicator requires additional description - how this indicator is constructed, the more that the cited source leads to the paper in Chinese.

4. Diatom indexes are crucial in this research, and in section 2.4.3. they are presented only in the form of enigmatic abbreviations. Each indicator requires at least a brief description, in particular where (for which region of the world) they were created and calibrated; necessarily with a reference to the literature source, where the full specification of each index can be read.

5. Conclusions, lines 400-404: 'The difference in environmental conditions between the area where the diatom index is created and the area where the diatom index is applied may affect the strength of the correlation between the diatom index and environmental factors. Therefore, in the process of water ecological status assessment, it is suggested to choose the diatom index developed in the area with little difference from the study area’;

This conclusion is poorly confirmed in the research results, without distinguishing the place of creation of individual indicators and comparing them with each other depending on the values obtained in the research object. This conclusion seems unjustified if the results have not been analyzed in this respect.

This also applies to the abstract, lines 25-28.

Additional comments

Line 18 (and 37, 55, 62 and more): The IBD index is called ‘BDI’ in its paper of origin. I suggest keeping the original abbreviation, to make it easier to find by internet search engines after the paper has been published.

Lines 136-137: The phrase ‘without considering the CODMn  concentration’ is used twice.

Lines 188, 189 and 192: Change ‘A. Minutissimum” to ‘A. minutissimum’ (small letter)

Line 226: ‘the concentration of nutrient oxygen (TN, NH3-N, TP)’ – the word 'oxygen' is probably a mistake here.

Line 293: The table has an incorrect title - it presents correlation coefficients between the values of diatom indexes and the results of water quality.

Line 361: Fig. 5. The numerical values of the correlation coefficients are too bright and hard to see. I understand that the brightness reflects the result, but nevertheless I suggest to darken the colors a bit.

Author Response

  1. Study area (lines 80-85) needs to be supplemented with climatic conditions (average annual temperatures, temperature of the coldest and hottest month of the year, annual precipitation). Due to the fact that the research was conducted in one season, meteorological data for the 2018 season should also be included here, with a reference to multi-year conditions. It is important whether the year of research was average or deviated from the average conditions, e.g. for the multi-year period 1991-2020. I am convinced that there are available data from weather stations in the study area.

Response: Thank you for your valuable comments. We have added hydrometeorological data.

  1. It is advisable to supplement the 'Material and methods' with a more precise characterization of the studied river and its catchment area: topography, nature of the river, use of the catchment area, degree of urbanization. The Tangwang River seems to be an excellent object for such research - the upper part of the river is very natural, with a poorly urbanized natural catchment, and the lower course is more developed. It is worth describing it in the content of the chapter.

Response: Thank you for your valuable comments. We have added data on the topography, geomorphology, and water system of the basin. 

  1. Subsection 2.3. (lines 101-110) needs to be supplemented with what laboratory methods were used to determine water quality indicators.

Also, the QHEI indicator requires additional description - how this indicator is constructed, the more that the cited source leads to the paper in Chinese.

Response: Thank you for your valuable comments. We have supplemented the laboratory methods and the detailed description of QHEI .

  1. Diatom indexes are crucial in this research, and in section 2.4.3. they are presented only in the form of enigmatic abbreviations. Each indicator requires at least a brief description, in particular where (for which region of the world) they were created and calibrated; necessarily with a reference to the literature source, where the full specification of each index can be read.

Response: We have added descriptions of all abbreviations for these indices, as well as introductory websites for these indices.

  1. Conclusions, lines 400-404: 'The difference in environmental conditions between the area where the diatom index is created and the area where the diatom index is applied may affect the strength of the correlation between the diatom index and environmental factors. Therefore, in the process of water ecological status assessment, it is suggested to choose the diatom index developed in the area with little difference from the study area’;

This conclusion is poorly confirmed in the research results, without distinguishing the place of creation of individual indicators and comparing them with each other depending on the values obtained in the research object. This conclusion seems unjustified if the results have not been analyzed in this respect.

This also applies to the abstract, lines 25-28.

Response: Thank you for your valuable comments. We have adjusted our conclusions. “Therefore, in the process of water ecological status assessment, it is suggested to select the diatom index which has a good correlation with the environmental factors in the study area or to build a new diatom index based on the diatom community and environmental elements in the study area.”

Additional comments

Line 18 (and 37, 55, 62 and more): The IBD index is called ‘BDI’ in its paper of origin. I suggest keeping the original abbreviation, to make it easier to find by internet search engines after the paper has been published.

Response: Thank you for your valuable comments. We have changed the “IBD” to “BDI”.

Lines 136-137: The phrase ‘without considering the CODMn  concentration’ is used twice.

Response: Thank you for your valuable comments. We have removed the duplicates.

Lines 188, 189 and 192: Change ‘A. Minutissimum” to ‘A. minutissimum’ (small letter)

Response: Thank you for your valuable comments. We have changed the “M” to “m”.

Line 226: ‘the concentration of nutrient oxygen (TN, NH3-N, TP)’ – the word 'oxygen' is probably a mistake here.

Response: Thank you for your valuable comments. We have changed the “the concentration of nutrient oxygen” to “the nutrient concentration”.

Line 293: The table has an incorrect title - it presents correlation coefficients between the values of diatom indexes and the results of water quality.

Response: Thank you for your valuable comments. We have changed the name of Table 2.

Line 361: Fig. 5. The numerical values of the correlation coefficients are too bright and hard to see. I understand that the brightness reflects the result, but nevertheless I suggest to darken the colors a bit.

Response: Thank you for your valuable comments. We have adjusted the font color in Fig. 5.

Reviewer 2 Report

Comments in the attachment.

Author Response

Lines 18-19: The abbreviation must be explained the first time it is used.

Response: Thank you for your valuable comments. We have added an explanation of the abbreviations

Line 22: “general” ?

Response: Thank you for your valuable comments, “general” has been changed to “moderate”

Site T02 is missing in Figure 1.

Response: Thank you for your valuable comments. We have made corrections to Figure 1.

Line 99: Please specify the manufacturer and country of origin of the equipment used.

Response: Thank you for your valuable comments. We have added the manufacturer and country of origin of the equipment used.

Lines 104-105: What was this "on-site pretreatment" ?

Response: Thank you for your valuable comments. In order to ensure the accuracy of sample measurement, some samples need to adjust the pH to prolong the storage time, which we call "on-site pretreatment".

Line 105: ammonium nitrogen (N-NH4+), Line 106: nitrate nitrogen (N-NO3-)

Response: Thank you for your valuable comments. We have made the correction.

Line 107: (bott.) or (Bott.) - Sometimes it is lowercase, sometimes it is uppercase.

Response: Thank you for your valuable comments. We uniformly changed it to Bott.

There is no information about the methods used to determine the nitrogen and phosphorus forms and with what equipment.

Response: Thank you for your valuable comments. We added the information of the experimental equipment.

Line 114: water body? river water is tested

Response: Thank you for your valuable comments, “Water body” has been changed to “water”.

Lines 129-130: aerobic organic substance ? will consume a large amount of oxygen ?

Response: Thank you for your valuable comments, “Aerobic organic substance” has been changed to “organic substance”.

Lines: 132, 133, 134, 136, 137 and more: concentrations of CODCr, CODMn -These parameters indirectly determine the overall concentration of organic matter in the water through oxygen demand but are not parameters that directly determine the concentration. In my opinion, it should be - values of CODCr, CODMn.

Response: Thank you for your valuable comments. In the revised draft, the description of CODCr and CODMn is uniformly changed to "value".

Lines 136-137: “therefore, without considering the CODMn concentration, in this study, without considering the CODMn concentration,…”

Response: Thank you for your valuable comments. We have deleted the duplicate part.

Lines 142-143: Only a few of these abbreviations are developed in the Introduction. It is necessary to arrange all the abbreviations of the indices used in the form of a table, together with the full name, year of publication, and the literature source in which the description of this index and the method of its calculation can be found.

Response: Thank you for your valuable comments. We have added descriptions of all abbreviations for these indices, as well as introductory websites for these indices.

Line 145: Please specify the program manufacturer and country of origin.

Response: Thank you for your valuable comments. We have specified the program manufacturer and country of origin

Line 147: 10 environmental factors ? below there are 11 environmental factors.

Response: Thank you for your valuable comments. In Table 2, 10 environmental factors were included in the correlation analysis.

Lines 168-170: Please specify the program manufacturer and country of origin.

Response: R is an open source software developed by Ross Ihaka and Robert Gentleman, we have specified the developer.

Line 195: Please define the terms: reference point and damaged point (lines 117, 162) or reference and damaged sites.

Response: Thank you for your valuable comments. We have changed the "reference point and damaged point" to “reference and damaged sites”.

Table 1. For the pH value, the average is not counted, but the range is given.

Response: Thank you for your valuable comments. We replaced it with the range of pH values.

Line 226: nutrient oxygen (TN, NH3-N, TP) ?

Response: We replaced it with the nutrient concentration

Lines 261-265: This snippet is suitable for introduction.

Response: We wanted to show that some of the diatom indices that performed well in this study also performed well in other areas.

Line 266: “…with environmental factors such as DO…” (only one factor was mentioned).

Response: Thank you for your valuable comments. IDSE is significantly correlated with environmental factors such as DO, COD, and biochemical oxygen demand.

Line 279: "general" ?

Response: We have changed the "general" to “moderate”.

Line 293: Is the description of Table 2 really correct ? The table contains the correlation coefficients.

Response: We have changed the title of Table 2 to ”correlation coefficients between the values of diatom indexes and the results of water quality”.

Line 300: “…in different lakes in the evaluation of Lake Erie,…”

Response: We have changed " different lakes " to “different areas”.

Lines 313-316: the correlation coefficients were also positively correlated ?

Response: Yes. In this study, the higher the correlation coefficient (absolute value) between diatom index and COD, the better the evaluation effect of diatom index in Tangwang River Basin.

Lines 321-325: It is redundant here. It is well known that K2Cr2O7 is a stronger oxidant than KMnO4.

Response: Thank you for your valuable comments. We have not deleted this part because we are worried that some scholars in the field of aquatic biology may not know so much about water quality testing methods. If you don't feel the need to keep it, we will remove it before publication.

Line 333: water body? river water is tested.

Response: We have changed "water body" to “water”.

Lines 337-339: “…to indicate the gradient of organic pollution caused by human activities, which should be the main reason for the positive correlation between the accuracy of diatom index evaluation and CODCr but negative correlation with CODMn.” – I am not convinced by this explanation. Explain how it is possible that an increase in the content of organic substances increases the accuracy in the case of using K2Cr2O7 as an oxidant, and in the case of using KMnO4 it decreases ?

Response: In Tangwang River Basin, the soil in the study area is rich in humus, and the concentration of humus in the river is high, so the humus in the river is easy to be oxidized when using the potassium permanganate method to determine the concentration of organic matter, which leads to the high CODMn value in the source area of the river which is less affected by human activities, and can not truly reflect the impact of human activities.  Owing to the influence of human activities, the amount of wastewater flowing into the lower reaches of Tangwang River has increased, and the content of industrial organic matter and macromolecular organic matter in the river has increased.  This part of organic matter is difficult to be oxidized when potassium permanganate is used for oxidation, but it is easier to be oxidized when potassium dichromate is used for oxidation.  Therefore, we infer that the CODCr value may better reflect the concentration of industrial organic matter and macromolecular organic matter discharged into the river by human activities in the process of water quality measurement in Tangwang River Basin.  That is, CODCr value can better indicate the degree of organic pollution caused by human activities, which should be the main reason for the positive correlation between the accuracy of diatom index evaluation and CODCr but negative correlation with CODMn.

Line 389: impaired group ? Is it the same as damaged group ?

Response: Thank you for your valuable comments. We have changed the "impaired group" to “damaged group”.

Line 393: "fair" or "general" - Please, specify what ecological status you are referring to.

Response: We have changed the "general" to “moderate”.
